# Pan-cancer landscape of protein kinase D3: An integrative TCGA multi-omics analysis of clinical, molecular, and immunological roles

**Jocshan Loaiza-Moss**[ID]**, Michael Leitges**[ID]*

Faculty of Medicine, Division of Biomedical Sciences, Memorial University of Newfoundland, St. John's, Newfoundland and Labrador, Canada

* mleitges@mun.ca

## Abstract

Cancer remains a leading cause of mortality worldwide and a significant barrier to improving quality of life across all populations. The protein kinase D family, including *PRKD3*, has been demonstrated to play a crucial role in cancer development through its involvement in regulating key cellular processes. Although growing evidence highlights the role of *PRKD3* in the tumorigenesis of certain cancers, a comprehensive pan-cancer analysis of *PRKD3* remains unavailable. To address this, we performed an integrative pan-cancer analysis of *PRKD3* using multi-omics datasets from The Cancer Genome Atlas, the Genotype-Tissue Expression project, and cBioPortal. We examined *PRKD3* expression, copy number variation, mutation, and DNA methylation, and evaluated their associations with clinicopathological features, patient survival, and diagnostic potential across 33 cancer types. Immune relevance was further assessed through correlations with immune infiltration, checkpoint gene expression, and immunotherapy response-related genomic biomarkers. Our results revealed that *PRKD3* expression was highly heterogeneous, showing significant upregulation in liver cancer, gastric cancer, and adrenocortical carcinoma, and downregulation in others. Elevated expression was consistently associated with poor prognosis and increased stromal, neutrophil, and cancer-associated fibroblast infiltration in adrenocortical carcinoma, liver cancer, and stomach cancer, whereas paradoxical associations with favorable outcomes were observed in kidney clear cell carcinoma. *PRKD3* expression also correlated with immune checkpoint molecules including PD-1, PD-L1, and CTLA-4, supporting an immunosuppressive role, while context-dependent associations with TMB and MSI highlighted its potential influence on tumor immunogenicity and responsiveness to immune checkpoint blockade. Collectively, these findings identify *PRKD3* as a potential context-dependent modulator of tumor biology, prognosis, and immune interactions, underscoring its potential as a biomarker of diagnostic, prognostic, and therapeutic relevance in precision oncology.

**Data availability statement:** All datasets analyzed in this study are publicly available from The Cancer Genome Atlas (TCGA, https://www.cancer.gov/tcga) and cBioPortal for Cancer Genomics (https://www.cbioportal.org/).

**Funding:** Canada Research Chairs: 950-232114 The funders had no role in study design, data collection and analysis, decision to publish, or preparation of the manuscript.

**Competing interests:** The authors have declared that no competing interests exist.

## Introduction

Despite significant advancements in therapeutic approaches, the global incidence and mortality rates of cancer continue to rise at an alarming pace. According to the World Health Organization, in 2022, approximately 20 million new cancer cases and 9.7 million cancer-related deaths were reported globally, and a 77% increase is projected for 2050 [1,2], which represents a major public health concern across the world. These statistics underscore the persistent global health burden of cancer and the critical need for improved strategies in early detection and treatment.

Protein kinases D (PKDs), a family of serine/threonine kinases, function as critical downstream effectors of Protein Kinase C and diacylglycerol signaling [3–5]. PKDs comprise three isoforms in humans: PKD1 (*PRKD1*, formerly PKCμ), PKD2 (*PRKD2*), and PKD3 (*PRKD3*, formerly PKCν) [6–8], which share a conserved structure composed of an N-terminal regulatory region with two cysteine-rich zinc finger domains and a pleckstrin homology domain, as well as C-terminal effector kinase domain [9,10]. PKDs are known to regulate distinct cellular processes such as differentiation, membrane trafficking, cytokinesis, and oxidative stress response [11–16], with certain functional redundancies [17,18]. In cancer development and progression, PKDs have been shown to play significant roles in proliferation, survival, epithelial-mesenchymal transition, migration, invasion, angiogenesis, and immune response [19,20]. More specifically, *PRKD3* has been reported to be differentially regulated in various malignancies, including colorectal, gastric, liver, prostate, and breast cancers [21–25]. Its expression has been associated with treatment efficacy and patient prognosis in hepatocellular carcinoma and invasive prostate cancer [24,26]. Despite these associations, its pan-cancer expression landscape, prognostic value, and links to immune modulation and therapy response remain uncharacterized. Therefore, its essential to delineate *PRKD3's* tumor type-specific roles, identify clinically actionable patterns, and evaluate its biomarker potential for precision oncology.

In this study, we conducted a comprehensive pan-cancer analysis of *PRKD3* using integrated TCGA, GTEx, and multi-omics datasets to characterize its expression patterns across tumor and normal tissues, evaluate associations with patient survival and tumor progression, and investigate genetic/epigenetic alterations with implications for immunoregulation and therapy response. We further assessed *PRKD3* relationships with tumor immune microenvironment, including stromal/immune cell infiltration and immune modulators, as well as predictors of immune checkpoint inhibitor efficacy such as tumor mutation burden (TMB) and microsatellite instability (MSI). Finally, we explored the potential relationship between *PRKD3* and the therapy efficiency in different immunotherapy syngeneic models. An overview of the workflow for our study is provided in S1 Fig, similar to approaches established in previous pan-cancer characterization studies [27–30].

## Materials and methods

### Data collection

Processed transcriptomic data for *PRKD3* across 33 cancer types (Log2 (TPM + 0.001) (transcripts per million)), along with corresponding clinical annotations,

were retrieved from The Cancer Genome Atlas (TCGA) [31,32], and normal tissue expression data from the Genotype-Tissue Expression (GTEx) project (https://www.gtexportal.org/) [33]. The TOIL TCGA/GTEX and PANCAN harmonized cohorts were accessed through the UCSC Xena platform (https://xenabrowser.net/datapages/, accessed November 2024) [34]. Data included samples from a range of tissue types, such as solid normal and tumor tissues, primary tumors, normal tissues, as well as blood-derived malignancies from bone marrow and peripheral blood. Furthermore, proteomic data across nine cancer types were obtained from the Clinical Proteomic Tumor Analysis Consortium (CPTAC) via the Cancer Proteogenomic Data Analysis Site (cProSite) (https://cprosite.ccr.cancer.gov/, accessed February 2025) [35,36].

The 33 cohorts encompassed: adrenocortical carcinoma (ACC), bladder urothelial carcinoma (BLCA), breast invasive carcinoma (BRCA), cervical squamous cell carcinoma and endocervical adenocarcinoma (CESC), cholangiocarcinoma (CHOL), colon adenocarcinoma (COAD), diffuse large B-cell lymphoma (DLBC), esophageal carcinoma (ESCA), glioblastoma multiforme (GBM), head and neck squamous cell carcinoma (HNSC), kidney chromophobe (KICH), kidney renal clear cell carcinoma (KIRC), kidney renal papillary cell carcinoma (KIRP), acute myeloid leukemia (LAML), brain lower grade glioma (LGG), liver hepatocellular carcinoma (LIHC), lung adenocarcinoma (LUAD), lung squamous cell carcinoma (LUSC), mesothelioma (MESO), ovarian serous cystadenocarcinoma (OV), pancreatic adenocarcinoma (PAAD), pheochromocytoma and paraganglioma (PCPG), prostate adenocarcinoma (PRAD), rectum adenocarcinoma (READ), sarcoma (SARC), skin melanoma (SKCM), stomach adenocarcinoma (STAD), testicular germ cell tumors (TGCT), thyroid carcinoma (THCA), thymoma (THYM), uterine corpus endometrial carcinoma (UCEC), uterine carcinosarcoma (UCS), and uveal melanoma (UVM).

### Differential expression and clinicopathological correlation analyses

Differential *PRKD3* expression between tumor and normal tissues was analyzed using both paired and unpaired samples from the TCGA and GTEx datasets. ComBat-batch correction was performed to mitigate non-biological variation between studies using the R package "sva" (v3.50), as described by Wang et al. [37]. Proteomic profiles across nine cancer types (BRCA, GBM, HNSC, LIHC, UAD, LUSC, OV, PAAD and UCEC) were retrieved from CPTAC datasets. Associations between *PRKD3* expression and clinicopathological features — including pathological stage, T stage, lymph node metastasis, and lymphatic invasion, per American Joint Committee on Cancer classification (AJCC), the International Federation of Gynecology and Obstetrics (FIGO), Masaoka, and Ann Arbor classifications — were evaluated via correlation analysis. Tumor stage relationships were confirmed using the GEPIA2.0 platform "Stage Plot" tool (http://gepia2.cancer-pku.cn/, accessed December 2024) [38]. Visualizations were performed in R with "ggplot2" (v3.5.2), excluding undefined stages [34].

### Diagnostic and prognostic analyses

To evaluate the diagnostic potential of *PRKD3*, Receiver Operating Characteristic (ROC) curves were generated for ComBat-corrected TCGA–GTEx pan-cancer cohorts using the R package "pROC" (v1.19) [39]. Diagnostic accuracy was assessed by the Area Under the Curve (AUC), with values > 0.7 considered indicative of robust predictive performance. For prognostic analysis, patient samples were stratified into high- and low-expression groups based on the median *PRKD3* expression level. Kaplan–Meier curves and univariate and multivariate Cox regressions, were performed to evaluate *PRKD3*'s association with overall survival (OS) and progression-free interval (PFI) across cancers using the R packages "survival" (v3.8-3) and "survminer" (v0.5.0) [40,41], following Liu et al.'s guidelines [42].

### Genetic and epigenetic alteration analysis

Genetic alteration data, including mutations and Copy Number variants (CNVs), were retrieved from TCGA cohorts via cBioPortal for Cancer Genomics network (https://www.cbioportal.org/, accessed December 2024) [43] to examine alteration–expression relationships. Additionally, the SMART database (http://www.bioinfo-zs.com/smartapp/, accessed in

March 2025) was used to assess associations between *PRKD3* expression and both global and site-specific DNA methylation status [44].

### Immune infiltration analysis

To characterize the relationship between *PRKD3* expression and the tumor immune microenvironment (TIME), we assessed the relationships with immune-related scores and immune cell infiltration patterns. Using the ESTIMATE algorithm, we derived associations between *PRKD3* and tumor purity metrics: Immune Score (immune cell abundance), Stromal Score (stromal cell infiltration), and Estimate Score (combined index) [45]. To further examine the relationships with specific immune cell infiltrations, we utilized the TIMER3.0 platform (https://compbio.cn/timer3/, accessed July 2025) [46], which integrates multiple computational frameworks, TIMER, MCP-counter, QuanTIseq, EPIC, and TIDE, for comprehensive immune cell deconvolutions. Finally, to address method discrepancies, we performed a Fisher-Z transformation meta-analysis of Spearman correlation coefficients to calculate pooled ρ and method heterogeneity and identify concordant immune infiltrate associations per cancer type.

### Association between expression and immune/immunotherapy response

To investigate the immunological role of *PRKD3*, we analyzed its association with previously reported immune modulators [47], including both inhibitory and stimulatory molecules, across cancer types. Its potential relevance for immunotherapy was further explored by examining correlations with tumor mutational burden (TMB) and microsatellite instability (MSI), where TMB was calculated as the number of nonsynonymous mutations normalized to a standardized exome size of 38 Mb [48], and MSI scores were retrieved from the TIMER database. Additionally, *PRKD3* expression was evaluated using the Tumor Immune Syngeneic Mouse (TISMO) database (http://tismo.cistrome.org/, accessed on July 2025) [49], which provides transcriptomic data from syngeneic mouse tumor models, including both *in vivo* profiles from immune checkpoint blockade (ICB)-treated tumors and *in vitro* profiles from cytokine-stimulated cancer cell lines. Expression patterns were compared between responders and non-responders before and after ICB therapy across multiple tumor models, as well as before and after cytokine stimulation in different cell lines, to elucidate the immunoregulatory function and potential therapeutic implications of *PRKD3*.

### Statistical analysis

R (v4.3.3) and associated packages were used for all statistical analyses. Differential expression comparisons were performed using Wilcoxon rank-sum. Clinicopathological, genetic, epigenetic, and immune correlations were assessed via Spearman's rank correlation. Multiple testing correction was performed using the Benjamini-Hochberg false discovery rate (FDR) procedure, with FDR < 0.05 considered statistically significant. Survival analyses employed Kaplan–Meier estimation and univariate Cox proportional hazards regression.

## Results

### *PRKD3* expression landscape in pan-cancer

To comprehensively assess differences in *PRKD3* expression across human cancers, we first analyzed mRNA levels between tumor and normal tissues using TCGA-only data. Fig 1A shows significantly higher *PRKD3* expression in tumor samples from CHOL, HNSC and LIHC, whereas cancers such as BRCA, KICH, KIRC, KIRP, LUAD, LUSC, PRAD, THCA, and UCEC exhibited lower expression compared to adjacent normal tissues. When comparing only TCGA paired samples, we observe a similar pattern in BRCA, CHOL, HNSC, KICH, LIHC, LUAD, PRAD, and THCA (S2 Fig). To extend these findings, we integrated TCGA data with the GTEx cohort, providing a broader and more robust comparative view of *PRKD3* expression across tumor and normal samples (Fig 1B). This analysis demonstrated elevated *PRKD3* mRNA expression in multiple tumor types, including batch-corrected ESCA, HNSC, LIHC, and STAD, and non-corrected GBM,

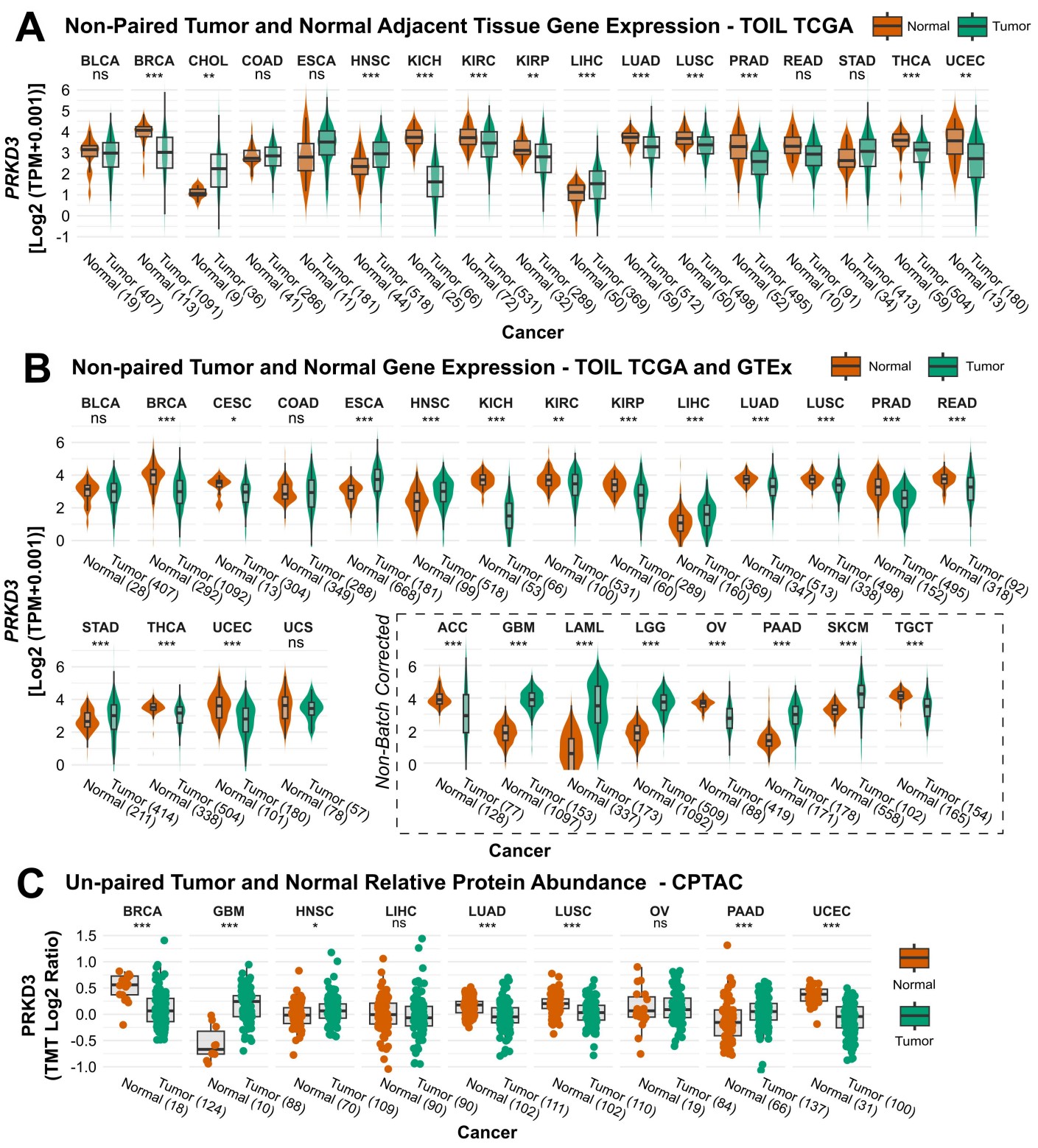

**Fig 1. Comparative Analysis of *PRKD3* Expression Across Pan-Cancer Cohorts.** (A) Differential *PRKD3* mRNA expression between tumor (Green) and adjacent normal tissues (Orange) based on TCGA-only data. (B) Differential *PRKD3* mRNA expression between unpaired tumor (Green) and normal (Orange) tissues using the batch-corrected TCGA–GTEx harmonized dataset. (C) Relative PRKD3 protein abundance (TMT log2 ratio) in

unpaired tumor (Green) versus normal (Orange) tissues from the TCGA–GTEx harmonized dataset. TMT: Tandem Mass Tag. Statistical significance was assessed using the Wilcoxon test and is indicated as follows: *FDR < 0.05, **FDR < 0.01, ***FDR < 0.001.

LAML, LGG, PAAD and SKCM. Conversely, reduced expression was observed in BRCA, CESC, KICH, KIRC, KIRP, LUAD, LUSC, PRAD, READ, THCA, and UCEC (batch-corrected), as well as ACC, OV, and TGCT (non-corrected). To validate these transcriptional findings at the protein level, we further examined PRKD3 expression using CPTAC data accessed via the cProSite database. Consistent with mRNA profiles, PRKD3 protein levels were significantly higher in GBM, HNSC, and PAAD tumors compared with normal tissues, but were reduced in BRCA, LUAD, LUSC, and UCEC tumors (Fig 1C). Collectively, these results reveal that *PRKD3* exhibits distinct and cancer-type–specific expression patterns, with broadly consistent trends at both the transcriptomic and proteomic levels.

## Clinicopathological relevance of *PRKD3* across human cancers

Given the distinct *PRKD3* expression patterns between tumor and normal tissues across cancers, we explored its association with various clinicopathological parameters. Using the GEPIA database, we examined *PRKD3* expression across different pathological stages in several cancers. Our analysis revealed significant stage-dependent variations in cancers such as ACC, KICH, LIHC, and STAD, where *PRKD3* expression generally increased in advanced stages. In contrast, KIRC and OV exhibited a significant decrease in *PRKD3* expression as the disease progressed (Fig 2A). A comprehensive comparison of *PRKD3* expression across all cancer stages is provided in S3 Fig.

Additionally, we conducted Spearman correlation analyses using AJCC classification pathological staging variables, including primary tumor size (T), lymph node involvement (N), and presence of metastasis (M) [50], along with other staging classification systems, such as Masaoka, Ann Arbor, and FIGO. Our analysis revealed that *PRKD3* expression positively correlated with pathological stage in ACC, LIHC, and STAD, but negatively correlated in KIRC, OV, and THYM (Fig 2B). TNM-specific staging variables showed positive correlations with T stage in LIHC and STAD, and negative correlations in KIRC and THCA. For N staging, positive correlations were observed in KIRP, STAD, and THCA, but negative correlations in CESC and TGCT; as well as a negative correlation with M stage in KIRC. These findings indicate a tumor type-specific association between *PRKD3* expression and clinicopathological variables, where higher expression associates with advanced disease in ACC, LIHC, and STAD, but with earlier-stage disease in KIRC, OV, and THYM. A complete overview of the clinicopathological distribution of the TCGA cohorts is provided in S1 Table.

## Prognostic and diagnostic relevance of *PRKD3* in pan-cancer

To assess the diagnostic potential of *PRKD3* across cancers, we performed ROC analyses using batch-corrected and non-corrected TCGA and GTEx datasets. ROC curves assess a biomarker's ability to distinguish tumor from normal tissues, where the AUC indicates predictive accuracy (AUC = 0.5: no discrimination; AUC = 1.0: perfect discrimination) [51]. *PRKD3* showed strong diagnostic performance (AUC > 0.7) in 17 cancer types with distinct patterns: elevated expression in CHOL and ESCA (batch-corrected) and GBM, LGG, LAML, PAAD, SKCM (non-corrected) enabled reliable tumor-normal discrimination, while reduced expression in KICH, BRCA, PRAD, UCEC, THCA, KIRP, LUAD (batch-corrected) and OV, TGCT, ACC (non-corrected) yielded similar accuracy (Fig 3A). Optimal cutoffs were determined using the maximal Youden index for each cancer, enabling accurate classification of tumor versus normal tissues (S2 Table). These findings highlight the cancer-specific diagnostic value of *PRKD3* and support its potential as a clinical biomarker.

Furthermore, we evaluated *PRKD3's* prognostic impact on cancer patient survival, including OS and PFI, via univariate and multivariate Cox regression and Kaplan–Meier analyses, stratifying cohorts into high- and low-expression groups based on the median *PRKD3* expression using the "survival" and "survminer" R packages. Univariate analyses revealed that elevated *PRKD3* expression was associated with increased OS hazard ratios (HR) in ACC (HR = 6.44), KICH

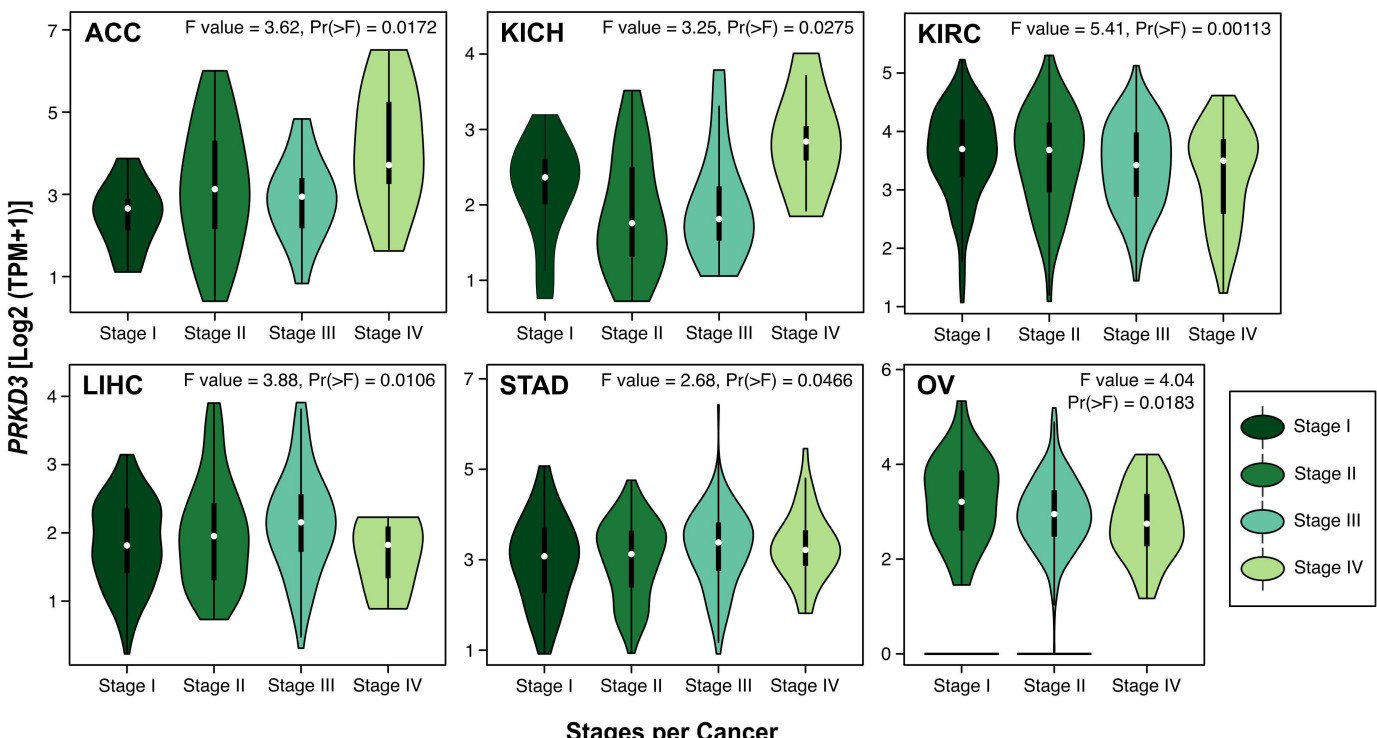

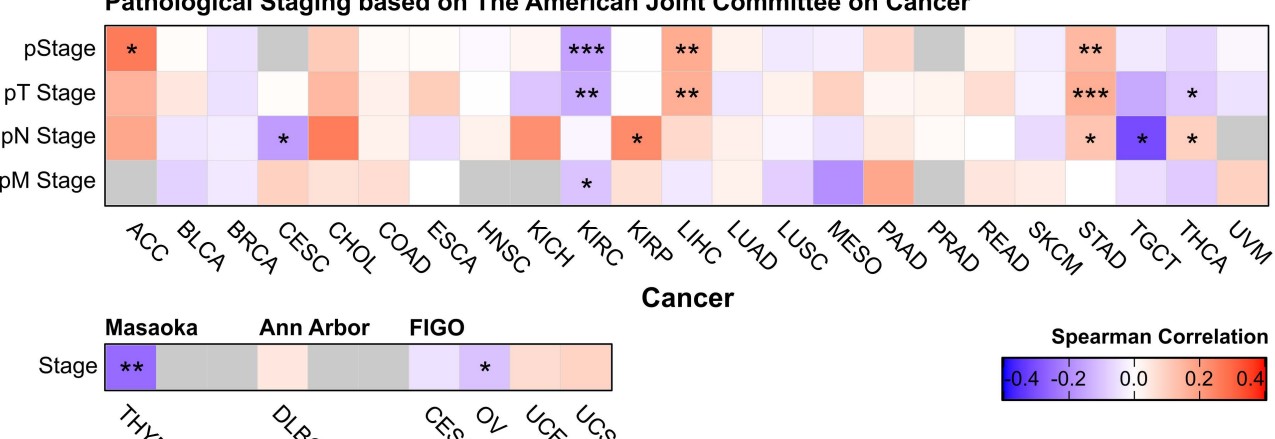

**Fig 2. Association Between *PRKD3* Expression and Clinicopathological Variables in Pan-Cancer.** (A) *PRKD3* expression across different cancer types and stages. Violin colors correspond to cancer stages. Data were obtained from the GEPIA database. (B) Correlations between *PRKD3* expression and clinicopathological variables, including Stage, T, N, and M, derived from TCGA GDC cohorts using distinct staging classifications. Box color denotes Spearman correlation coefficients (red representing positive and blue represents negative correlations). Statistical significance is indicated as *FDR < 0.05, **FDR < 0.01, ***FDR < 0.001.

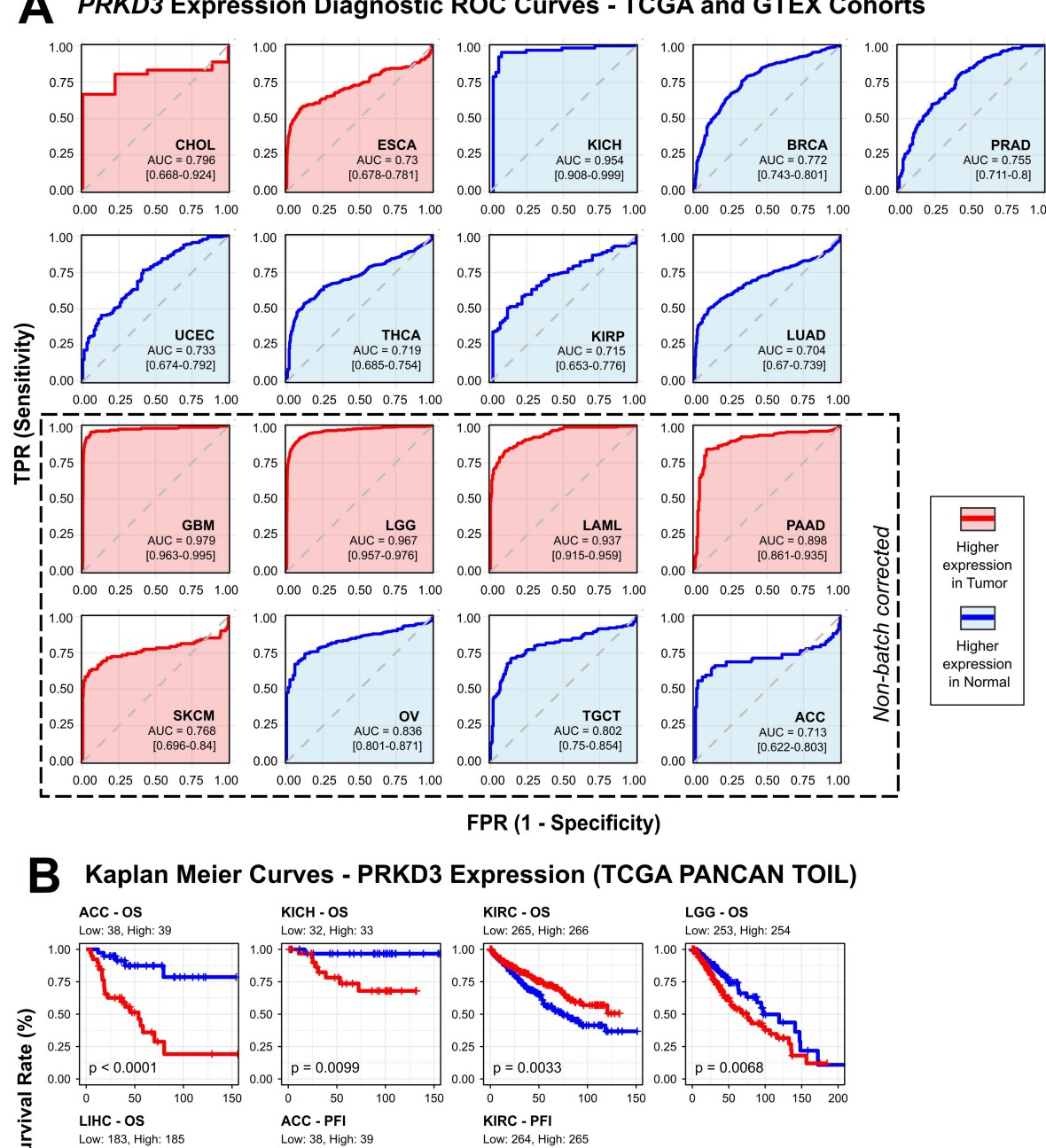

**Fig 3. Diagnostic and Prognostic Value of *PRKD3* Expression in Pan-Cancer.** (A) Receiver operating characteristic (ROC) curves for cancer types with AUC > 0.7, based on TCGA and GTEx datasets. Red curves represent cancers in which *PRKD3* expression is higher in tumor versus normal tissue, whereas blue curves indicate lower *PRKD3* expression in tumors. (B) Kaplan–Meier survival curves for overall survival (OS) and progression-free

interval (PFI) in TCGA cancer cohorts, using a median-based stratification of *PRKD3* expression. Red curves correspond to high *PRKD3* expression and blue curves to low *PRKD3* expression; only cohorts with p < 0.05 are shown.

(HR = 9.45), LGG (HR = 1.66) and LIHC (HR = 1.48), but reduced HR in KIRC (HR = 0.63) (S3 Table). PFI associations included elevated HR in ACC (HR = 4.74) and reduced HR in KICH (HR = 0.67). Kaplan–Meier analyses further illustrated shorter survival in ACC (OS, PFI), KICH (OS), LGG (OS) and LIHC (OS), versus improved survival in KIRC (OS, PFI) (Fig 3B). Additionally, multivariate Cox regressions adjusting for sex, race, age and pathological stage showed independent OS associations with elevated HR in ACC (HR = 4.04), KICH (HR = 14.7), LGG (HR = 1.61), LIHC (HR = 1.54), STAD (HR = 1.45), but reduced HR in KIRC (HR = 0.71) and LUAD (HR = 0.71). Multivariate PFI analyses showed elevated HR in ACC (HR = 3.91) and LGG (HR = 1.25) (S4 Table). These results indicate a cancer type-specific prognostic impact of *PRKD3*, where high expression is linked to poorer outcomes in ACC, KICH, LGG, LIHC and STAD, but appears protective in KIRC and LUAD. Overall, our findings highlight the complex role of *PRKD3* as both a potential diagnostic and prognostic biomarker across diverse cancer types.

## Association of *PRKD3* expression with genetic and epigenetic alterations

To investigate potential mechanisms underlying *PRKD3* dysregulation, we examined its association with genetic and epigenetic alterations across cancers. Genetic aberrations, including mutations and CNVs are known to affect gene function and transcriptional activity, contributing to tumor heterogeneity and oncogenic progression [52]. Using the cBioPortal platform, we analyzed *PRKD3*'s mutational landscape across 10,036 cases, revealing a low overall somatic mutation frequency (0.96%; 128 events: 112 missense, 10 truncating, and 6 splice-site), with highest frequencies in UCEC (4.5%, 510 cases), UCS (3.5%, 57 cases), READ (3.2%, 154 cases), SKCM (2.8%, 467 cases) and COAD (2.6%, 427 cases) (Fig 4A). However, *PRKD3* expression did not differ significantly between mutated and wild-type tumors pan-cancer-wide (Fig 4B), suggesting short mutations are unlikely a major driver of *PRKD3* expression. Subsequently, analyzed *PRKD3* CNVs using GISTIC-derived TCGA data across 10,749 cases, revealing a low overall alteration frequency (0.6%) dominated by amplifications. Peak amplification rates occurred in TGCT (2.7%, 150 cases), LUAD (2.2%, 507 cases), DLBC (2.1%, 48 cases), LUSC (2.0%, 492 cases), and BLCA (1.8%, 391 cases), whereas deep deletions were rare in KICH (1.5%, 66 cases) and THCA (0.2%, 498 cases) (Fig 4C). Notably, our analysis revealed amplifications were significantly associated with elevated *PRKD3* expression in BRCA, LUAD, OV, and TGCT (Fig 4D). These findings indicate that CNVs, in particular amplifications, represent a potential contributor to *PRKD3* upregulation in specific malignancies, while point mutations play a minor role.

To explore the role of epigenetic regulation in *PRKD3* dysregulation and tumorigenesis, we analyzed DNA methylation patterns using TCGA datasets via the SMART online platform, identifying 11 methylation probes in CpG islands, N-shore, N-shelf, and open sea regions (Fig 5A). Pan-cancer CpG-aggregated methylation was elevated in BRCA, CHOL, COAD, KIRC, LUAD and LUSC, but reduced in BLCA, LIHC, and PRAD (Fig 5B). To assess functional implications, we next examined correlations between site-specific *PRKD3* methylation and gene expression using Spearman analysis (Fig 5C). CpG island–associated sites showed negative correlations in ACC, BRCA, CHOL, DLBC, ESCA, LAML, and STAD. Open sea regions exhibited more heterogeneous relationships with *PRKD3* expression: negative correlations were detected in ACC, SKCM, and TGCT, but positive in BLCA, BRCA, GBM, HNSC, KIRP, LGG, PAAD, PRAD, and STAD. Methylation in N Shore regions generally showed negative correlations with *PRKD3* expression across most cancer types. These findings highlight that *PRKD3* expression may be differentially regulated by DNA methylation across tumor types, suggesting that epigenetic modifications contribute to its tumor-specific expression patterns. Importantly, distinct cancer-specific methylation patterns were identified, including the CpG island hypomethylation coupled with open sea hypermethylation, suggesting regional methylation interplay may also exert opposing regulatory effects on *PRKD3* transcription depending on the cancer context.

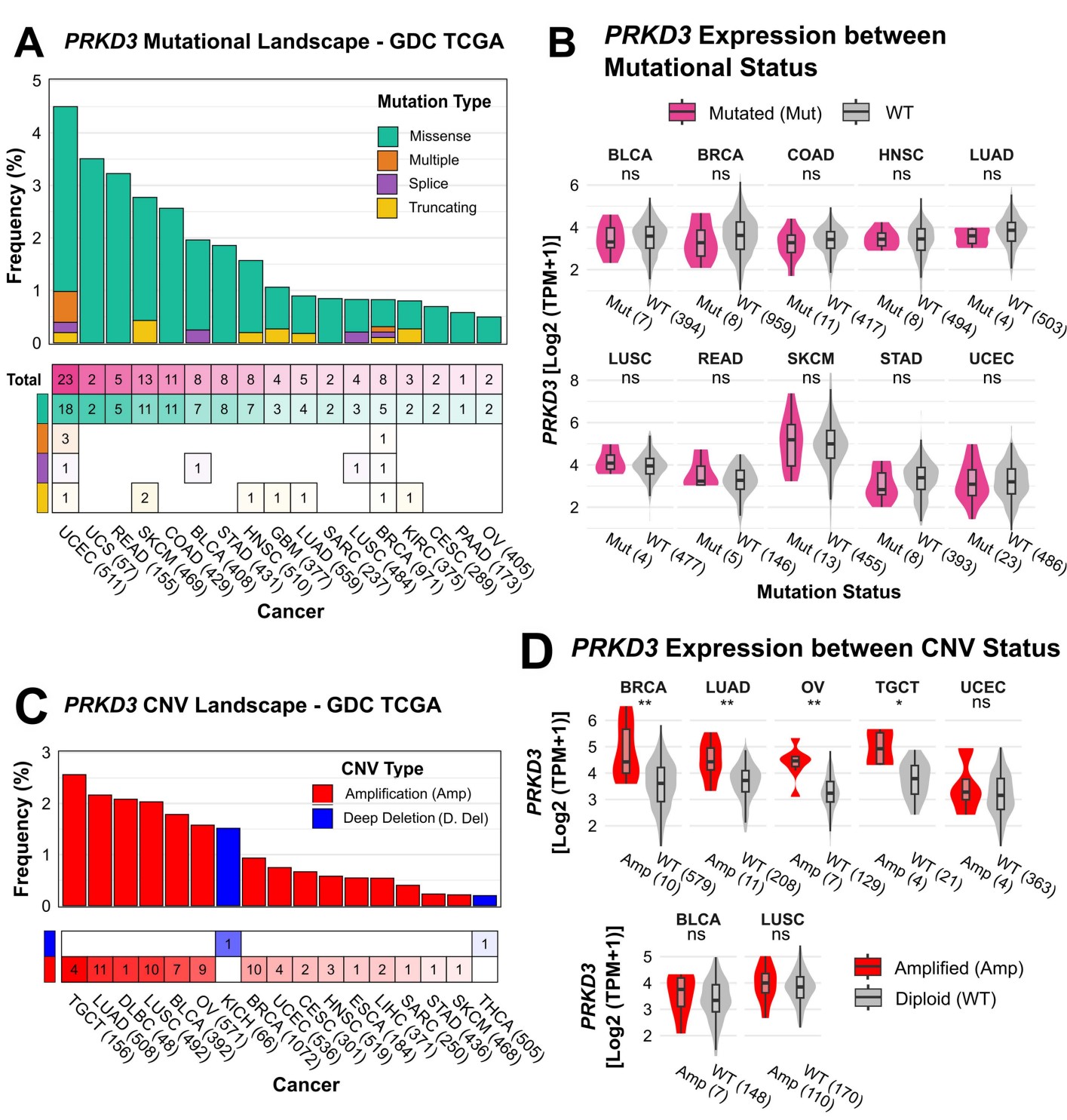

**Fig 4. Genetic Alteration Analysis and Association with *PRKD3* Expression in Pan-Cancer.** (A) Mutational landscape of *PRKD3* across cancers by mutation type and event frequency. Numbers inside boxes indicate the number of reported events per cancer, with colors representing mutation types. (B) Comparison of *PRKD3* expression between mutant (pink) and wild-type (gray) samples (n > 4). (C) *PRKD3* copy number variation (CNV) frequency landscape across cancers. Numbers inside boxes indicate the number of reported events per cancer. CNV types include amplifications (red) and deep deletions (blue). (D) Comparison of *PRKD3* expression between samples with amplifications (red) and diploid status (gray) (n > 4). Statistical significance was assessed using the Wilcoxon test (*FDR < 0.05, **FDR < 0.01, ***FDR < 0.001).

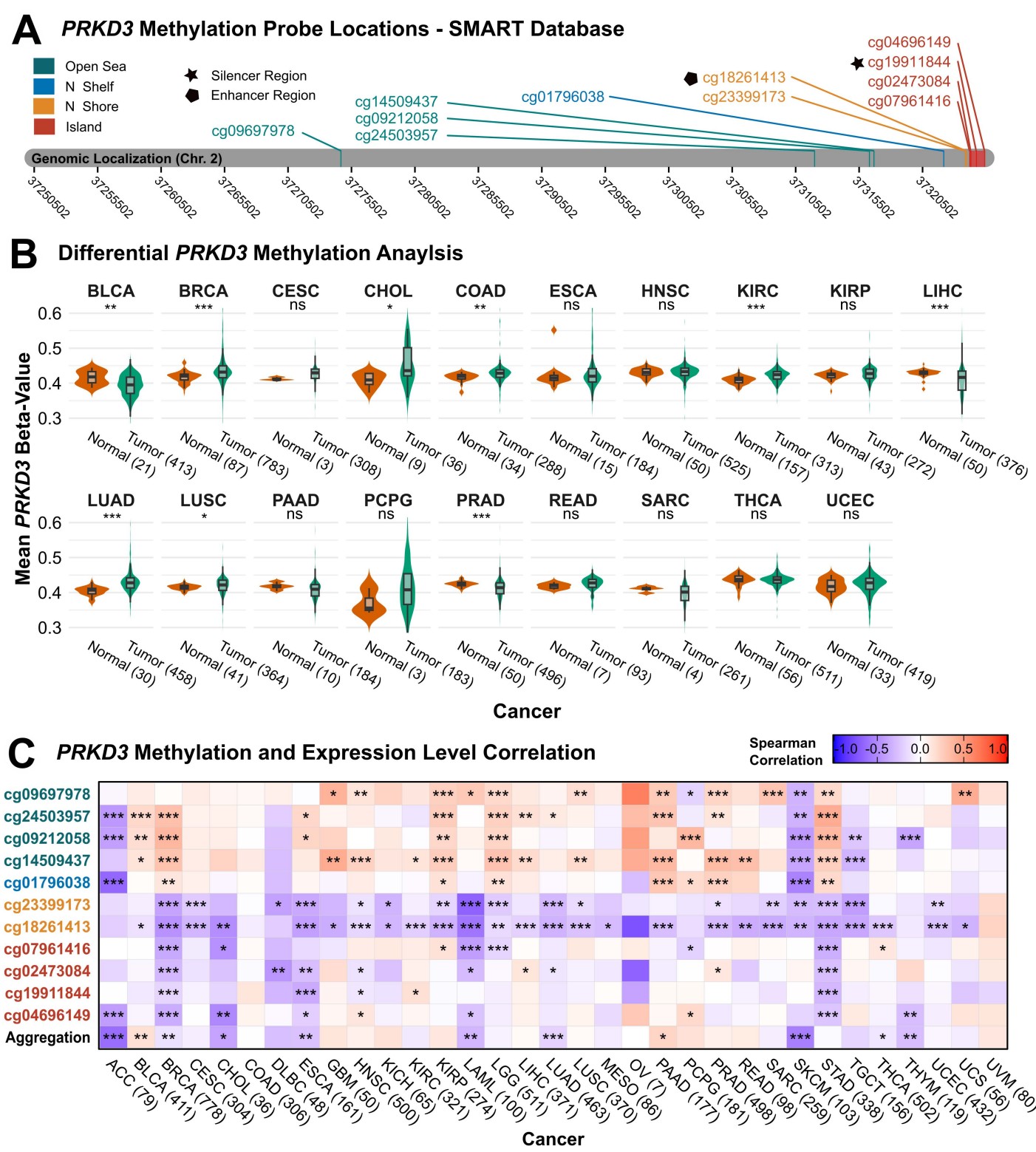

**Fig 5. *PRKD3* Methylation Patterns and Association with Gene Expression in Pan-Cancer.** (A) Genomic distribution of *PRKD3*-associated methylation probes. Colors and labels indicate distinct genomic regions linked to the *PRKD3* gene. (B) Comparison of mean *PRKD3* methylation (beta-values) between tumor (green) and normal (orange) tissue samples (n > 4). (C) Correlation between *PRKD3* expression and site-specific methylation (mean

beta-values) across cancers. Box color denotes Spearman correlation coefficients, with red representing positive and blue representing negative correlations. Statistical significance is indicated as *FDR<0.05, **FDR<0.01, ***FDR<0.001.

## Immune infiltration patterns associated with *PRKD3* expression

To examine the relationship between *PRKD3* expression and the TIME, we performed tumor purity-adjusted correlations with ESTIMATE-derived Immune, Stromal, and Estimate Scores across cancers. We observed positive correlations with all three scores in BRCA, HNSC, LGG, LUAD, LUSC, PAAD, and STAD, indicating that higher *PRKD3* levels are associated with increased immune and stromal cell infiltration (Fig 6A), while negative correlations Immune and Estimate Scores correlations were observed for the in CESC, SARC, and UCEC, suggesting *PRKD3* association with reduced immune cell infiltration in these cancers. Overall, these results indicate that *PRKD3* may exert a context-dependent effect on the tumor immune microenvironment across different cancer types.

To further dissect these associations to specific immune cell populations, we employed multiple computational frameworks integrated within TIMER3.0 (TIMER, MCP-counter, QuanTIseq, EPIC, and TIDE) (S4 Fig). Notably, HNSC, LIHC, PAAD, PRAD, STAD, and THCA exhibited significant associations with several immune cell populations, suggesting a broad immunoregulatory effect in these tumor contexts, while ACC, CHOL, GBM, SKCM, and UCS showed relatively few significant associations, which may reflect either a weaker immunological role of *PRKD3* or greater tumor-intrinsic heterogeneity. To address method-associated correlation discrepancies, we performed a meta-analysis on the significant coefficients to identify concordant infiltrate associations (consensus ≥3 methods). Overall, *PRKD3* showed robust positive correlation with low heterogeneity for neutrophil infiltration, T cell CD8+ and cancer-associated fibroblast (CAF) abundance across multiple tumor types, highlighting its potential role in regulating both immune and stromal components of the tumor microenvironment (Fig 6B).

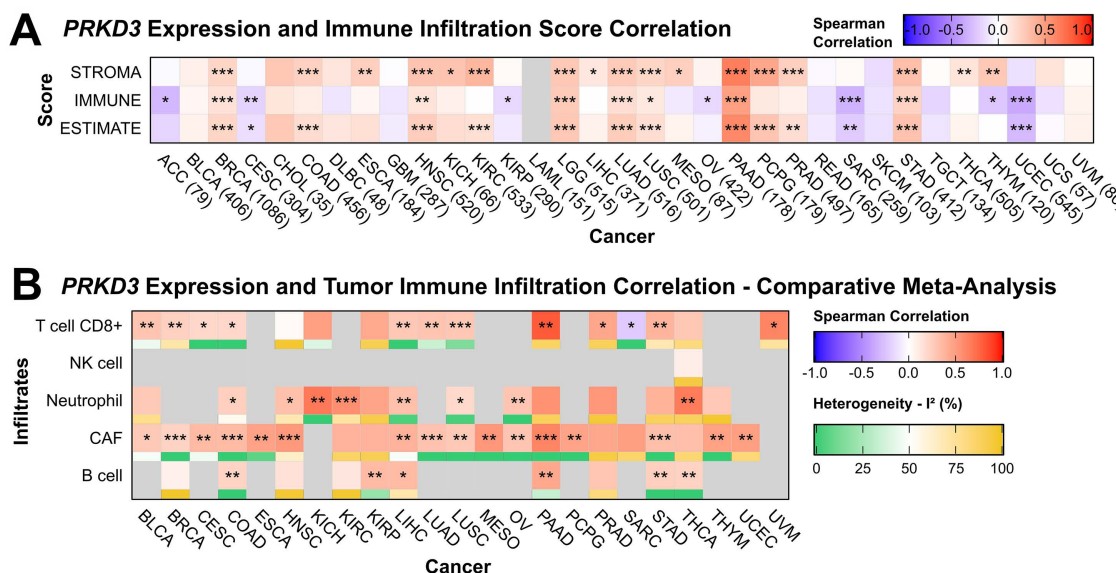

**Fig 6. Association Between *PRKD3* Expression and Immune Infiltration in Pan-Cancer.** (A) Correlation between *PRKD3* expression and ESTI-MATE infiltration scores, including Immune, Stromal, and ESTIMATE scores. (B) Meta-analysis of Spearman correlation coefficients for a consensus of ≥3 algorithms via Fisher Z-transformation: pooled ρ and method heterogeneity (I²). Box color represents Spearman correlation coefficients, with red indicating positive correlations and blue indicating negative correlations. Statistical significance is indicated as: *FDR<0.05, **FDR<0.01, ***FDR<0.001.

## Immune landscape and immunotherapy response associated with *PRKD3* expression

To investigate the potential role of *PRKD3* in modulating the TIME and influencing response to immunotherapy, we analyzed correlations between *PRKD3* expression and immune-related features across cancers. Given that the therapeutic efficacy of immune checkpoint inhibitors (ICIs) is tightly dependent on the TIME [53], we examined the co-expression patterns between *PRKD3* and a panel of 60 immune checkpoint pathway genes, both immunostimulatory and immunoinhibitory molecules, using Spearman correlations across TCGA cohorts. Most checkpoint genes were significantly co-expressed with *PRKD3*, with the majority showing positive associations, including key inhibitory molecules such as PD-L1 and CTLA-4 (Fig 7), suggesting a potential role for *PRKD3* in modulating immune evasion mechanisms. However, cancer-specific exceptions were observed, with tumors including ACC, GBM, SARC, SKCM, and UCS exhibiting weak or negative correlations. Additionally, we examined the association of *PRKD3* with genomic features reported to influence ICI efficacy, tumor mutational burden (TMB) and microsatellite instability (MSI), which serve as predictive biomarkers of immunotherapy response. Correlation analyses across TCGA cohorts revealed that *PRKD3* expression was positively associated with TMB in ACC, BRCA, LAML, and LIHC, but negatively correlated with TMB in ESCA, KIRP, PAAD, PRAD, STAD, and THCA (Fig 8A). Similarly, *PRKD3* expression was positively correlated with MSI in KIRP and PAAD, whereas negative correlations were observed in BLCA, BRCA, COAD, HNSC, LGG, OV, STAD, and THYM. Collectively, these findings suggest that *PRKD3* expression is intricately linked to multiple layers of immune regulation across cancers, including immune checkpoint pathways, and may also interact with genomic instability in a cancer-type–specific manner, potentially modulating tumor immunogenicity and influencing responsiveness to ICI therapy.

To further evaluate the functional relevance of *PRKD3* to immunotherapy response, we analyzed *in vivo* mouse syngeneic treatment models and *in vitro* cytokine stimulation assays from public datasets integrated through the TISMO platform. *In vivo* treatment studies showed that higher *PRKD3* expression in breast and gastric tumors was associated with increased sensitivity to anti-PD-L1 and anti-CTLA4/anti-PD-1 combination therapies, respectively, while lower *PRKD3* expression correlated with sensitivity to the anti-CTLA-4/anti-PD-1 combination in breast and colorectal tumor models (Fig 8B). *In vitro* cytokine stimulation experiments revealed *PRKD3* downregulation in response to IFN-γ treatment in skin, pancreatic, and lung tumor models, and IFN-β treatment in lung cancer models (Fig 8C). Across both experimental settings, these findings suggest that *PRKD3* may influence tumor immune responsiveness through modulation of the tumor microenvironment and immune signaling pathways in syngeneic models.

## Discussion

The complexity and heterogeneity of cancer pose significant challenges to effective clinical management and therapeutic intervention. A comprehensive understanding of the molecular and cellular mechanisms driving carcinogenesis is critical for the development of targeted and personalized treatment strategies. Protein kinases, including *PRKD3*, serve as central hubs of oncogenic signaling and drug responsiveness through the regulation of proliferation, cell cycle progression, and survival pathways. While *PRKD3* dysregulation has been documented in multiple malignancies, prior evidence largely derives from cell line models [21,54–56], limiting our understanding of its functional impact in a clinical context.

Advances in high-throughput sequencing, including resources such as TCGA and GTEx, provide extensive pan-cancer expression data that allow for comprehensive analyses of normal and tumor tissues. Leveraging these datasets, our analysis revealed that, relative to normal tissues, *PRKD3* expression was significantly elevated in several cancers, including CHOL, ESCA, GBM, HNSC, LAML, LGG, LIHC, PAAD, SKCM, and STAD. Consistent with these findings, previous studies have reported elevated *PRKD3* in LIHC [26], STAD [21], and oral squamous cell carcinoma (OSCC) [57], one of the predominant subtypes of HNSC. These results indicate that *PRKD3* may contribute to tumor initiation and progression in a cancer-dependent manner. *PRKD3* was also downregulated in multiple other cancers, including ACC, BRCA, KICH, KIRC, KIRP, LUAD, LUSC, OV, PRAD, THCA, and UCEC. These observations highlight a potential dual role for *PRKD3* as a context-dependent oncogenic or tumor-suppressive role across cancer lineages, emphasizing the need for mechanistic

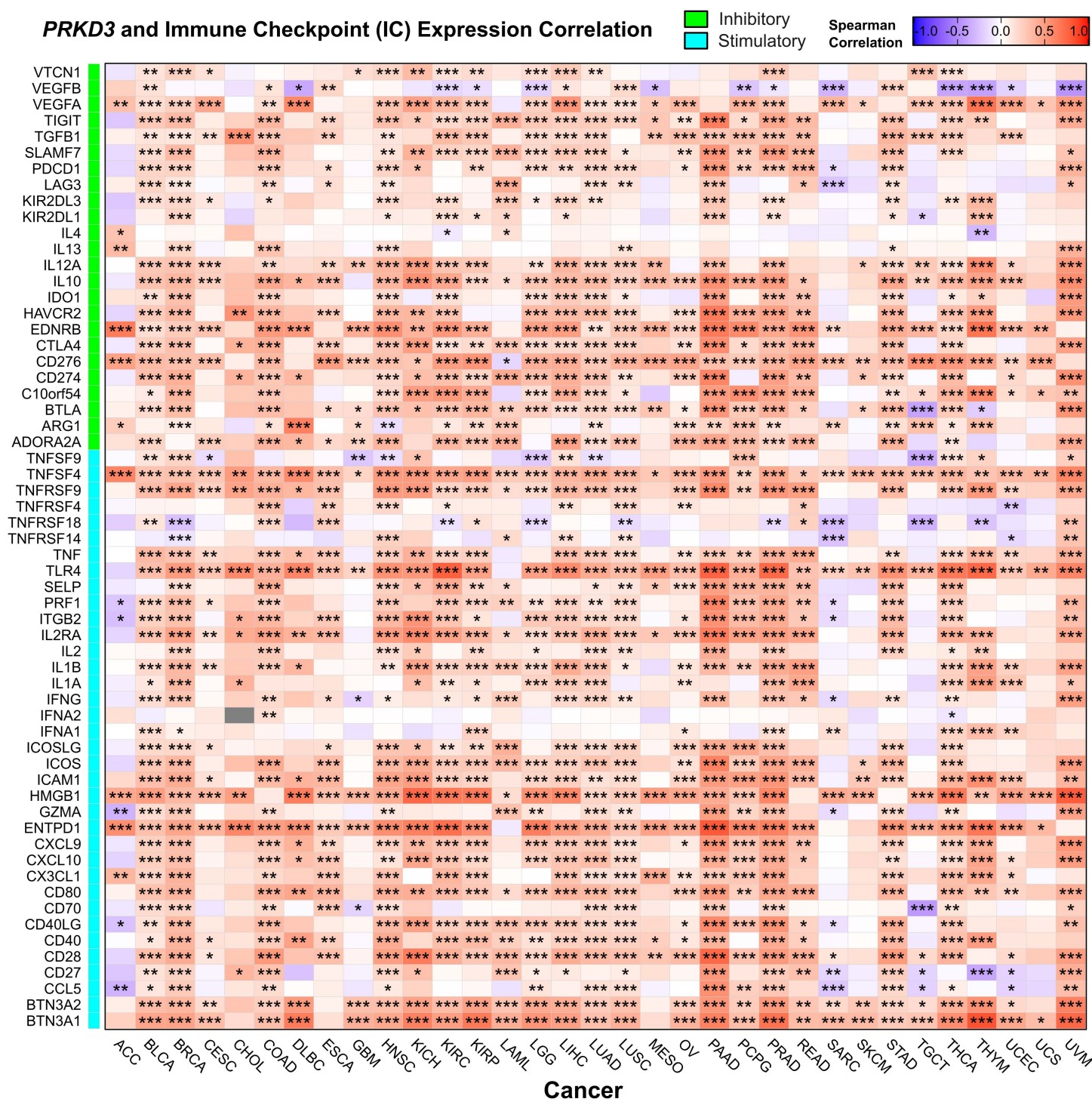

**Fig 7. Co-Expression Association Between *PRKD3* and Several Immune Checkpoint Molecules.** Label colors indicate inhibitory (green) and stimulatory (cyan) effects. Box color represents the Spearman correlation coefficient, with red indicating positive correlations and blue indicating negative correlations. Statistical significance is indicated as follows: *FDR<0.05, **FDR<0.01, ***FDR<0.001.

## A Correlation between *PRKD3* Expression and Tumor Genomic Heterogeneity Biomarkers

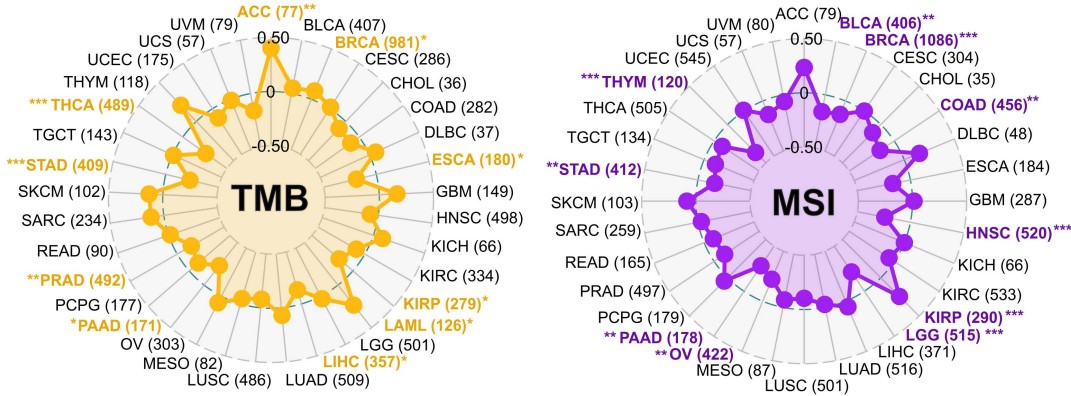

## B *PRKD3* Expression and *In Vivo* IC Blockade Treament

Baseline  Non−responders  Responders

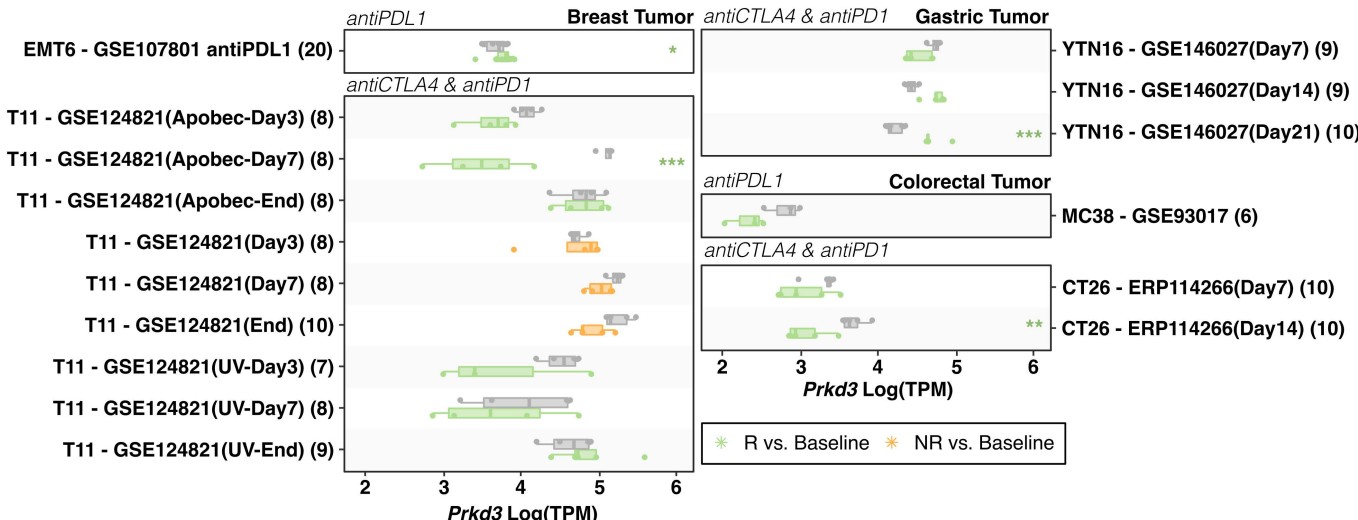

## C *PRKD3* Expression and *In Vitro* Cytokine Treament

Baseline  IFNb  IFNg  TNFa

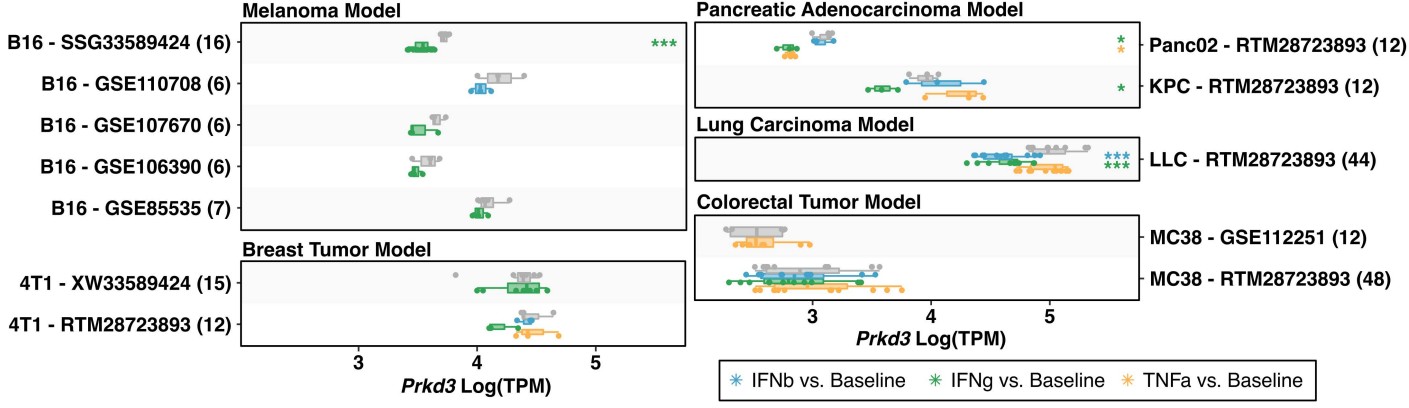

**Fig 8. Association Between *PRKD3* Expression and Immunotherapy Sensitivity.** (A) Correlation between *PRKD3* expression and genomic biomarkers of tumor heterogeneity. Tumor mutation burden (TMB) is shown in yellow, and microsatellite instability (MSI) in purple. Statistical significance is highlighted in the cancer labels as follows: *FDR<0.05, **FDR<0.01, ***FDR<0.001. (B) *In vivo* analysis of syngeneic tumor models treated with immunotherapy,

comparing responders (green), non-responders (yellow), and baseline (gray) with respect to *PRKD3* expression. Data were obtained from the TISMO database. (C) *In vitro* analysis of cytokine-treated tumor models with respect to *PRKD3* expression. Data were obtained from the TISMO database.

studies to elucidate its functional contribution to tumorigenesis. Additionally, considering the established link between gene expression and disease progression [58,59], our analyses revealed context-specific associations between *PRKD3* and pathological stage and TNM classification: positive correlations in ACC, LIHC, and STAD, but negative correlations in KIRC, OV, and THYM.

Several differential *PRKD3* expression patterns translated into prognostic and diagnostic significance. Our Kaplan–Meier and Cox regression analyses (univariate and multivariate) demonstrated that high *PRKD3* expression is strongly associated with poor prognosis in ACC, KICH, LGG, and LIHC, and STAD, consistent with prior reports in LIHC and STAD [26,60,61], where it may serve as independent prognostic indicator. Notably, our study, to knowledge, is the first to report contexts where elevated *PRKD3* correlates with favorable outcomes, particularly in KICH, highlighting its context-dependent role of in tumor progression. Additionally, our findings suggest that *PRKD3* may also have diagnostic utility, allowing to discriminate tumor tissues from normal counterparts in cancer-specific contexts. An intriguing observation from our analysis is that BRCA exhibits reduced *PRKD3* expression relative to normal tissue, suggesting its potential as a diagnostic biomarker. However, previous studies using estrogen receptor–negative and triple-negative breast cancer cell lines reported *PRKD3* upregulation in tumors [62–64]. Given limited prior data on *PRKD3* clinicopathological associations and expression discrepancies, further clinical studies across diverse patient cohorts and molecular subtypes are required to validate the potential links reported in this study.

Our analyses revealed a consistent association between elevated *PRKD3*, tumor development, and poor prognosis in LGG, LIHC, and STAD, whereas KIRC exhibited the opposite pattern. Intriguingly, *PRKD3* also displayed paradoxical behavior in ACC and KICH, where reduced expression was associated with tumor diagnosis, yet higher intra-tumoral expression correlated with disease progression and worse outcomes. To our knowledge, *PRKD3* has not previously been characterized as a paradoxical gene; hence, we hypothesize a temporal expression dynamic involving initial suppression during tumor initiation, followed by intra-tumoral upregulation that drives aggressive progression. These cancer type-specific patterns suggest *PRKD3* functions as a potential context-dependent oncogene, with prognostic significance determined by intra-tumoral expression gradients rather than absolute tumor-normal differences. However, future studies adjusting for tumor purity, immune subtypes, and molecular classifications will be essential to validate these observations and elucidate the underlying biological mechanisms.

Genetic and epigenetic alterations are fundamental drivers of tumorigenesis, influencing transcriptional regulation and thereby enhancing or suppressing protein activity, which contributes to cancer progression [65]. In the case of *PRKD3*, these alterations remain uncharacterized. In our pan-cancer analysis, *PRKD3* primarily exhibited low-frequency SNVs and CNVs, with amplifications being the most prevalent and significantly correlated with elevated expression in BRCA, LUAD, OV, TGCT. In addition to these findings, DNA methylation analyses, comprising 11 probes located in CpG islands, N-shore, N-shelf, and open sea regions, revealed pan-cancer hypermethylation in BRCA, CHOL, KIRC, LUAD, and LUSC, but hypomethylation in BLCA, LIHC, and PRAD. Notably, methylation within CpG island probes, putatively overlapping a silencer element, showed inverse correlations with *PRKD3* expression in several cancers, consistent with repressive effects on transcription, but a positive association in KIRC. In contrast, N-shore probes, which encompass an enhancer-like region, generally displayed negative methylation–expression correlations, suggesting that hypomethylation at these sites could maintain enhancer activity and support *PRKD3* transcription. Open sea sites exhibited more heterogeneous relationships, compatible with context-dependent effects on distal regulatory elements. Together, these data support a model in which site-specific methylation across silencer- and enhancer-enriched regions may exert opposing influences on *PRKD3* expression, contributing to its tumor- and context-specific dysregulation.

Another important aspect of our study is the role of *PRKD3* in modulating the TIME and its potential association with immunotherapy sensitivity. *PRKD3* expression was broadly correlated with stromal infiltration across most cancers, suggesting a role in microenvironment remodeling, while immune infiltration associations were cancer-specific: positive correlations in BRCA, HNSC, PAAD, and STAD, and negative correlations in ACC, SARC, and UCEC. Taking a deeper look into the specific infiltrating populations, we found that high *PRKD3* expression was consistently associated with increased levels of neutrophils and CAFs, two key players known to drive stromal remodeling, immune evasion, and immunotherapy resistance, consistent with a pro-tumorigenic and immunosuppressive phenotype, as well as limited checkpoint inhibitor efficacy [66,67]. Importantly, while associations with neutrophils, CD8+T cells and CAFs were consistent across deconvolution algorithms, other immune cells varied methodologically, highlighting the need for cross-platform validation to ensure robust correlations.

We further evaluated *PRKD3*'s potential role of in modulating tumor immunogenicity through co-expression with immune checkpoint genes and associations with mutational biomarkers, which may influence immunotherapy response [47]. *PRKD3* expression showed positive correlations with major inhibitory immune checkpoints, including PD-1, PD-L1, and CTLA-4 across cancers, suggesting a potential role in immune evasion. *PRKD3* also exhibited context-dependent associations with genomic immunogenicity markers, including MSI and TMB [68]. Negative MSI correlations in BLCA, COAD, HNSC, LGG, OV, and THYM suggest an association with reduced mismatch repair–driven neoantigen generation. TMB showed positive correlations in ACC, LAML, and LIHC, indicating that high *PRKD3* coincides with elevated mutational burden and neoantigen production independent of MSI, but negative associations in ESCA, PRAD, and THCA, reflecting lower intrinsic immunogenicity. Considering the complex interplay between these genomic factors [69], MSI and TMB also exhibited diverse co-occurrence patterns across cancers, suggesting *PRKD3's* potential context-dependent immunogenicity modulation: (a) low MSI/TMB profiles indicate limited neoantigen burden despite high *PRKD3* expression; (b) high MSI/low TMB suggests mismatch repair defects as the primary neoantigen driver; and (c) low MSI/high TMB demonstrates compensatory mutational load promoting immune recognition. These patterns highlight how *PRKD3* may influence tumor immunogenicity and immunotherapy responsiveness through cancer-specific MSI–TMB landscapes and microenvironmental interactions.

To extend these observations, we analyzed TISMO mouse datasets to evaluate *PRKD3*'s potential relationship with therapeutic outcomes in checkpoint blockade and cytokine-based treatments. *In vivo* data showed that higher *PRKD3* expression was associated with responders to anti-CTLA-4/anti-PD-1 therapy in gastric cancer, potentially reflecting elevated CD4+T cells and cytotoxic lymphocytes, while lower expression correlated with responders in colorectal cancer, consistent to *PRKD3*'s inverse MSI relationship. *In vitro*, IFN-γ and IFN-β treatment reduced *PRKD3* expression in melanoma, lung, and pancreatic tumor models, suggesting that inflammatory cytokine signaling can modulate *PRKD3* levels and influence immune responsiveness. These observations align with functional evidence in OSCC, where *PRKD3* regulates PD-L1 via STAT1/STAT3 signaling, supporting a mechanistic role in immune evasion and PD-1/PD-L1 therapy efficacy [70]. Although derived from mouse syngeneic models that differ from human tumors, these data suggest that *PRKD3* may modulate immune responsiveness through microenvironmental and signaling alterations, indicating cancer-specific predictive value as a potential immunotherapy biomarker that warrants further human clinical validation.

Our study has several limitations that should be acknowledged. First, all analyses were conducted using publicly available datasets without external cohorts or additional experimental or clinical validation, which limits causal inference and the generalizability of our findings. Moreover, methylation–expression relationships were not systematically adjusted for copy number variation, which may act as an important confounder in interpreting epigenetic regulation of *PRKD3*. The immunotherapy-related analyses also rely on mouse syngeneic models and *in vitro* systems, which only partially represent human tumor immunobiology and therefore require cautious extrapolation to clinical contexts. Future studies could incorporate experimental models such as orthotopic organoids in NOD-SCID mice [71] or multicellular 3D spheroids [72], which would allow targeted perturbation of *PRKD3* and its pathways in more physiologically relevant contexts,

providing mechanistic insight and functional validation. Finally, although this study integrates multi-omics layers including transcriptomics, DNA methylation, and proteomics, additional regulatory dimensions such as *PRKD3* transcript variants and post-translational modifications were not explored and may critically influence its activity and downstream signaling. Addressing these gaps is essential to determine whether *PRKD3* functions as a driver of tumor biology and immunomodulation, and to clarify its therapeutic relevance across cancer types.

In conclusion, our study provides a comprehensive pan-cancer overview of *PRKD3*, suggesting a potential role as a context-dependent modulator of tumor biology, prognosis, and immune responsiveness. High expression consistently associated with poor outcomes and increased stromal/ neutrophil infiltration in ACC, LIHC, and STAD, whereas KIRC exhibited paradoxical associations. *PRKD3* also correlated with key immune checkpoints and MSI/TMB status across cancers, alongside cancer-specific patterns in TISMO immunotherapy datasets, highlighting its potential role in shaping the TIME and influencing immunotherapy response. Overall, our findings propose *PRKD3* as a potential context-dependent biomarker to inform precision oncology; however, further validation studies are needed to establish mechanistic roles and clinical utility across diverse tumor types.

## Supporting information

**S1 Fig. Workflow for the pan-cancer characterization of *PRKD3*.** Schematic overview of the integrated analytical pipeline used to characterize *PRKD3* across cancers.
(TIFF)

**S2 Fig. Differential *PRKD3* mRNA expression between paired tumor (Green) and adjacent normal tissues (Orange) based on TCGA-only data.** Statistical significance was assessed using the Wilcoxon test and is indicated as follows: *FDR<0.05, **FDR<0.01, ***FDR<0.001.
(TIFF)

**S3 Fig. Pan-cancer comparative analysis of *PRKD3* expression across pathological stages.** Different shades of green represent distinct cancer stages. Data were obtained from the GEPIA2.0 platform.
(TIFF)

**S4 Fig. Analysis of immune cell infiltration using EPIC, MCP-counter, QuanTIseq, TIMER, and TIDE algorithms.** Box color represents Spearman correlation coefficients, with red indicating positive correlations and blue indicating negative correlations. Statistical significance is indicated as: *FDR<0.05, **FDR<0.01, ***FDR<0.001.
(TIFF)

**S1 Table. Clinicopathological characteristics of the pan-cancer cohorts included in this study.**
(XLSX)

**S2 Table. Diagnostic Receiver Operating Characteristic (ROC) curve analysis of PRKD3 expression across cancer types.** AUC, area under the curve; CI, confidence interval.
(XLSX)

**S3 Table. Hazard ratios (HR) for overall survival (OS) and progression free interval (PFI) based on PRKD3 expression across pan-cancer cohorts.** CI: Confidence Interval. KM: Kaplan-Meier p-value.
(XLSX)

**S4 Table. Pan-Cancer Multivariate Cox Regression for Overall Survival and Progression-Free Interval.** HR: Hazard Ratio; CI: Confidence Interval; BAA: Black or African American.
(XLSX)

## Author contributions

**Conceptualization:** Jocshan Loaiza-Moss, Michael Leitges.

**Data curation:** Jocshan Loaiza-Moss.

**Formal analysis:** Jocshan Loaiza-Moss.

**Funding acquisition:** Michael Leitges.

**Investigation:** Jocshan Loaiza-Moss.

**Methodology:** Jocshan Loaiza-Moss.

**Project administration:** Michael Leitges.

**Resources:** Michael Leitges.

**Supervision:** Michael Leitges.

**Validation:** Michael Leitges.

**Visualization:** Jocshan Loaiza-Moss.

**Writing – original draft:** Jocshan Loaiza-Moss.

**Writing – review & editing:** Jocshan Loaiza-Moss, Michael Leitges.

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
