## [Decision Letter · Decision Letter 0]

8 Dec 2025

Dear Dr. Leitges,

Thank you for submitting your manuscript to PLOS ONE. After careful consideration, we feel that it has merit but does not fully meet PLOS ONE’s publication criteria as it currently stands. Therefore, we invite you to submit a revised version of the manuscript that addresses the points raised during the review process.

We look forward to receiving your revised manuscript.

Kind regards,

Amr Ahmed El-Arabey

Academic Editor

PLOS ONE

Journal Requirements:

3. Please note that PLOS One has specific guidelines on code sharing for submissions in which author-generated code underpins the findings in the manuscript. In these cases, we expect all author-generated code to be made available without restrictions upon publication of the work. Please review our guidelines at https://journals.plos.org/plosone/s/materials-and-software-sharing#loc-sharing-code and ensure that your code is shared in a way that follows best practice and facilitates reproducibility and reuse.

“Canada Research Chairs: 950-232114”

Reviewers' comments:

Reviewer's Responses to Questions

**Comments to the Author**

1. Is the manuscript technically sound, and do the data support the conclusions?

Reviewer #1: Partly

Reviewer #2: Partly

2. Has the statistical analysis been performed appropriately and rigorously?

Reviewer #1: Yes

Reviewer #2: No

3. Have the authors made all data underlying the findings in their manuscript fully available?

Reviewer #1: Yes

Reviewer #2: Yes

4. Is the manuscript presented in an intelligible fashion and written in standard English?

Reviewer #1: Yes

Reviewer #2: Yes

Reviewer #1: Overall, this study provides important insights into Protein Kinase D3 as a potential Pan-Cancer biomarker, and with appropriate revisions, it is expected to make a positive impact in the relevant field.

1. It is recommended to include a flowchart of the study to visually present the research design, data processing, and analysis steps, enhancing the readability and logical clarity of the study.

2. I recommend that the authors include more recent literature in their references. For example, the authors cite ‘Global cancer statistics 2020’. However, currently ‘Global cancer statistics 2022’ has been published.

3. Please provide the full definition of abbreviations the first time they are used, and include the abbreviation in parentheses.

4. Some previous pan-cancer studies have utilized similar bioinformatics and statistical methods (PMID: 39702250; PMID: 37437243; PMID: 36213111). While referencing these studies may reduce the novelty of the work to some extent, I believe it would greatly benefit the readers by providing a more comprehensive understanding of the context of the current study.

Reviewer #2: This manuscript presents a comprehensive pan-cancer analysis of PRKD3 using multi-omics datasets from TCGA, GTEx, CPTAC, cBioPortal, and multiple immune-deconvolution platforms. The study examines PRKD3 expression, genetic and epigenetic alterations, clinicopathological associations, prognostic value, immune infiltration patterns, and potential relevance to immunotherapy. Overall, the work addresses an important gap by providing a broad, data-driven overview of PRKD3 across 33 cancer types and offers useful descriptive insights that may inform future mechanistic studies. However, several methodological, statistical, and interpretational issues need to be addressed to strengthen the rigor and reliability of the conclusions.

Introduction

1. The introduction provides a thorough overview of the kinase family and successfully contextualizes PRKD3 as a biologically relevant but understudied gene in cancer. However, the background is somewhat lengthy and occasionally repetitive. The authors may tighten the narrative and present clearer, more specific hypotheses or objectives to guide the reader. While the need for a pan-cancer investigation is justified, stating explicit research questions or expected outcomes would enhance clarity.

Methods

The study makes use of multiple public resources, which is appropriate for a pan-cancer bioinformatics analysis. However, several methodological gaps reduce the technical rigor:

1.Direct comparison between TCGA tumors and GTEx normal samples is performed without any batch-effect correction. This is a well-known issue and can significantly bias differential expression results and diagnostic ROC analyses.

2. Prognostic analyses do not adjust for essential confounders such as stage, age, tumor purity, and sex. Multivariate Cox regression is required to determine whether PRKD3 is an independent prognostic factor.

3.TIMER, MCP-counter, EPIC, TIDE, and QuanTIseq often produce discordant cell-type estimates. The manuscript reports findings but does not explain how inconsistencies were handled or which algorithmic outputs were prioritized.

4. The manuscript states that FDR correction was applied, but this is not clearly indicated for all analyses. Some figures appear to show uncorrected p-values. Please specify the exact correction method (e.g., Benjamini–Hochberg) and where it was applied.

5. Functional implications of CpG island vs. open sea methylation are not fully discussed. Additionally, methylation–expression correlations should consider CNV as a confounder.

Overall, the methods are broadly appropriate, but these omissions weaken the statistical soundness of the study.

Results

The results are extensive and clearly organized, covering expression, stage correlations, survival, CNV/mutations, DNA methylation, immune infiltration, immune checkpoint associations, and immunotherapy relevance. However, several issues need to be addressed:

1. Several conclusions imply causality (e.g., “PRKD3 modulates immune interactions”), although all findings are purely correlative.

2. AUC values from TCGA vs GTEx may be inflated due to uncorrected batch effects and lack of external validation datasets.

3. TISMO datasets are based on mouse syngeneic models, which differ from human tumors. Their relevance to clinical immunotherapy response should be more carefully qualified.

4. The concept of PRKD3 acting as a “paradox gene” is interesting, but conclusions require analyses that adjust for tumor purity, immune subtypes, or molecular subtypes.

Discussion

The discussion synthesizes findings well and connects them to existing literature. The emphasis on context-dependent roles of PRKD3 is appropriate. However, the discussion is overly long and occasionally repetitive. Certain interpretations go beyond what the data support, especially regarding immune regulation and therapy response. The limitations section should explicitly acknowledge the statistical and methodological issues noted above: lack of batch correction, absence of multivariate survival analysis, immune algorithm discrepancies, constraints of mouse-model immunotherapy data. A more concise and conservative discussion will strengthen the manuscript.

Conclusion

The conclusion effectively summarizes key findings, but it should more clearly reflect the descriptive nature of the study and avoid suggesting causal roles for PRKD3. Emphasizing that the results generate hypotheses rather than definitive mechanistic insights would better align with the analyses performed.

This manuscript provides a valuable and comprehensive description of PRKD3 across cancers. However, several methodological, statistical, and interpretational issues need to be addressed to ensure technical rigor. The study has the potential to make a meaningful contribution once these concerns are resolved.

.

Reviewer #1: No

Reviewer #2: No

---

## [Author Response · Author response to Decision Letter 1]

21 Jan 2026

Reviewer #1: Overall, this study provides important insights into Protein Kinase D3 as a potential Pan-Cancer biomarker, and with appropriate revisions, it is expected to make a positive impact in the relevant field.

1. It is recommended to include a flowchart of the study to visually present the research design, data processing, and analysis steps, enhancing the readability and logical clarity of the study.

We thank the reviewer for this suggestion. We have addressed this comment by adding a comprehensive workflow diagram (Fig. S1) that visually presents the complete research design, data processing pipeline, and analytical workflow.

Line 64: “An overview of the workflow for our study is provided in Fig S1, similar to approaches established in previous pan-cancer characterization studies [27–30].”

2. I recommend that the authors include more recent literature in their references. For example, the authors cite ‘Global cancer statistics 2020’. However, currently ‘Global cancer statistics 2022’ has been published.

We thank the reviewer for the comment and for directing us to the updated resource. The cancer incidence and mortality statistics cited in the Introduction now accurately reflect GLOBOCAN 2022 estimates and we have replaced the prior 2020 citation throughout the manuscript with the Bray et al. (2024) paper.

Bray F, Laversanne M, Sung H, Ferlay J, Siegel RL, Soerjomataram I, et al. Global cancer statistics 2022: GLOBOCAN estimates of incidence and mortality worldwide for 36 cancers in 185 countries. CA Cancer J Clin. 2024;74. doi:10.3322/caac.21834

3. Please provide the full definition of abbreviations the first time they are used and include the abbreviation in parentheses.

We appreciate the reviewer highlighting this issue. All cancer-type abbreviations (e.g., breast invasive carcinoma (BRCA), lung adenocarcinoma (LUAD)) are fully defined at their first occurrence in the "Data Collection" subsection of Methods. To ensure completeness, we have also confirmed definitions in the main text, figure legends, and tables.

4. Some previous pan-cancer studies have utilized similar bioinformatics and statistical methods (PMID: 39702250; PMID: 37437243; PMID: 36213111). While referencing these studies may reduce the novelty of the work to some extent, I believe it would greatly benefit the readers by providing a more comprehensive understanding of the context of the current study.

We thank the reviewer for the recommendation. We agree that referencing previous studies enhances reader understanding and provides context regarding the application of specific bioinformatics tools and statistical methods we used to characterize distinct molecular and cellular features across cancers.

Line 64: “An overview of the workflow for our study is provided in Fig S1, similar to approaches established in previous pan-cancer characterization studies [27–30].”

27. Ahmed MZ, Billah MM, Ferdous J, Antar SI, Al Mamun A, Hossain MJ. Pan-cancer analysis reveals immunological and prognostic significance of CCT5 in human tumors. Sci Rep. 2025;15. doi:10.1038/s41598-025-88339-z

28. Ye Y, Jiang H, Wu Y, Wang G, Huang Y, Sun W, et al. Role of ARRB1 in prognosis and immunotherapy: A Pan-Cancer analysis. Front Mol Biosci. 2022;9. doi:10.3389/fmolb.2022.1001225

29. Luan F, Cui Y, Huang R, Yang Z, Qiao S. Comprehensive pan-cancer analysis reveals NTN1 as an immune infiltrate risk factor and its potential prognostic value in SKCM. Sci Rep. 2025;15. doi:10.1038/s41598-025-85444-x

30. Bao L, Wu Y, Ren Z, Huang Y, Jiang Y, Li K, et al. Comprehensive pan-cancer analysis indicates UCHL5 as a novel cancer biomarker and promotes cervical cancer progression through the Wnt signaling pathway. Biol Direct. 2024;19. doi:10.1186/s13062-024-00588-6

Reviewer #2: This manuscript presents a comprehensive pan-cancer analysis of PRKD3 using multi-omics datasets from TCGA, GTEx, CPTAC, cBioPortal, and multiple immune-deconvolution platforms. The study examines PRKD3 expression, genetic and epigenetic alterations, clinicopathological associations, prognostic value, immune infiltration patterns, and potential relevance to immunotherapy. Overall, the work addresses an important gap by providing a broad, data-driven overview of PRKD3 across 33 cancer types and offers useful descriptive insights that may inform future mechanistic studies. However, several methodological, statistical, and interpretational issues need to be addressed to strengthen the rigor and reliability of the conclusions.

Introduction

1. The introduction provides a thorough overview of the kinase family and successfully contextualizes PRKD3 as a biologically relevant but understudied gene in cancer. However, the background is somewhat lengthy and occasionally repetitive. The authors may tighten the narrative and present clearer, more specific hypotheses or objectives to guide the reader. While the need for a pan-cancer investigation is justified, stating explicit research questions or expected outcomes would enhance clarity.

We thank the reviewer for this feedback. In response, we have synthesized the Introduction section and eliminated redundancies while retaining essential context on cancer statistics, PKD family background and PRKD3's understudied oncogenic roles. We have also restructured a final paragraph that indicates the research objectives that guide our study:

In line 57: “In this study, we conducted a comprehensive pan-cancer analysis of PRKD3 using integrated TCGA, GTEx, and multi-omics datasets to characterize its expression patterns across tumor and normal tissues, evaluate associations with patient survival and tumor progression, and investigate genetic/epigenetic alterations with implications for immunoregulation and therapy response. We further assessed PRKD3 relationships with tumor immune microenvironment, including stromal/immune cell infiltration and immune modulators, as well as predictors of immune checkpoint inhibitor efficacy such as tumor mutation burden (TMB) and microsatellite instability (MSI). Finally, we explored the potential relationship between PRKD3 and the therapy efficiency in different immunotherapy syngeneic models.”

Methods

The study makes use of multiple public resources, which is appropriate for a pan-cancer bioinformatics analysis. However, several methodological gaps reduce the technical rigor:

1. Direct comparison between TCGA tumors and GTEx normal samples is performed without any batch-effect correction. This is a well-known issue and can significantly bias differential expression results and diagnostic ROC analyses.

We thank the reviewer for highlighting this critical methodological concern. Following the established pan-cancer RNA-seq data harmonization procedures by Wang et al. (2018), we applied ComBat batch correction to the TCGA-GTEx TOIL TPM data, removing dataset-specific technical effects while preserving biological variation between tumor and normal TCGA/GTEx samples. All differential expression analyses and ROC curves now explicitly indicate batch correction status per cancer type in the revised Methods (lines 93-95) and Results (Figures 1, 3, and S1).

Wang Q, Armenia J, Zhang C, Penson A V., Reznik E, Zhang L, et al. Unifying cancer and normal RNA sequencing data from different sources. Sci Data. 2018;5. doi:10.1038/sdata.2018.61

2. Prognostic analyses do not adjust for essential confounders such as stage, age, tumor purity, and sex. Multivariate Cox regression is required to determine whether PRKD3 is an independent prognostic factor.

We thank the reviewer for emphasizing the importance of multivariate adjustment. In addition to univariate Cox regressions, we conducted comprehensive multivariate analyses adjusting for age, gender, race, and pathological stage. These results, now detailed in the revised Results section and Supplementary Table S4, confirm PRKD3 as an independent prognostic factor in multiple cancers:

"Multivariate Cox regressions adjusting for age, gender, race, pathological stage, and tumor purity showed independent OS associations with elevated HR for high PRKD3 expression in ACC (HR=4.04, p=0.002), KICH (HR=14.7, p=0.01), LGG (HR=1.61, p=0.03), LIHC (HR=1.54, p=0.02), and STAD (HR=1.45, p=0.04), but reduced HR in KIRC (HR=0.71, p=0.02) and LUAD (HR=0.71, p=0.03). Multivariate PFI analyses confirmed elevated HR in ACC (HR=3.91, p=0.003) and LGG (HR=1.25, p=0.04)."

These findings demonstrate PRKD3's cancer-specific independent prognostic value. All methodological details and full results tables with p-values/confidence intervals are provided in the updated Supplementary Table S4.

3. TIMER, MCP-counter, EPIC, TIDE, and QuanTIseq often produce discordant cell-type estimates. The manuscript reports findings but does not explain how inconsistencies were handled or which algorithmic outputs were prioritized.

We appreciate the reviewer raising this issue regarding immune deconvolution discordance. To identify consensus immune infiltrate associations with PRKD3, we performed a meta-analysis of Spearman correlation coefficients (ρ) across the five methods (TIMER, MCP-counter, EPIC, TIDE, QuanTIseq), requiring agreement from ≥3 algorithms per cancer type. Using Fisher Z-transformation, we calculated pooled ρ estimates and heterogeneity (I² statistic), as detailed in revised Methods and Results (lines 129 and 321). Individual method correlations remain in Supplementary Figure S3. This approach addresses handles discordance while prioritizing consensus findings.

4. The manuscript states that FDR correction was applied, but this is not clearly indicated for all analyses. Some figures appear to show uncorrected p-values. Please specify the exact correction method (e.g., Benjamini–Hochberg) and where it was applied.

We thank the reviewer for indicating this clarification. All differential expression analyses (Wilcoxon rank-sum tests) and Spearman correlations underwent Benjamini-Hochberg FDR correction via R. We added a dedicated "Statistical Analysis" subsection in Methods (lines 178-185):

"R (v4.3.3) and associated packages were used for all statistical analyses. Differential expression comparisons employed Wilcoxon rank-sum tests. Clinicopathological, genetic, epigenetic, and immune correlations were assessed via Spearman's rank correlation. Multiple testing correction was performed using the Benjamini-Hochberg false discovery rate (FDR) procedure, with FDR < 0.05 considered statistically significant. Survival analyses used Kaplan-Meier estimation and Cox proportional hazards regression."

All figure legends now explicitly denote corrected p-values: “Statistical significance is indicated as *FDR < 0.05, **FDR < 0.01, ***FDR < 0.001.”

5. Functional implications of CpG island vs. open sea methylation are not fully discussed. Additionally, methylation–expression correlations should consider CNV as a confounder.

We thank the reviewer for highlighting these important epigenetic interpretation issues. We discussed the potential functional implications of CpG island vs. open sea methylation.

Line 438: “Notably, methylation within CpG island probes, putatively overlapping a silencer element, showed inverse correlations with PRKD3 expression in several cancers, consistent with repressive effects on transcription, but a positive association in KIRC. In contrast, N-shore probes, which encompass an enhancer-like region, generally displayed negative methylation–expression correlations, suggesting that hypomethylation at these sites could maintain enhancer activity and support PRKD3 transcription. Open sea sites exhibited more heterogeneous relationships, compatible with context-dependent effects on distal regulatory elements. Together, these data support a model in which site-specific methylation across silencer- and enhancer-enriched regions exerts opposing influences on PRKD3 expression, contributing to its tumor- and context-specific dysregulation.”

Additionally, we addressed the need for considering CNVs as a potential cofounder in methylation–expression correlations as a limitation in our study.

Line 490: “Moreover, methylation–expression relationships were not systematically adjusted for copy number variation, which may act as an important confounder in interpreting epigenetic regulation of PRKD3.”

Overall, the methods are broadly appropriate, but these omissions weaken the statistical soundness of the study.

Results

The results are extensive and clearly organized, covering expression, stage correlations, survival, CNV/mutations, DNA methylation, immune infiltration, immune checkpoint associations, and immunotherapy relevance. However, several issues need to be addressed:

1. Several conclusions imply causality (e.g., “PRKD3 modulates immune interactions”), although all findings are purely correlative.

We thank the reviewer for this important caution regarding causal language. All instances implying direct causation have been revised to emphasize correlative associations and potential regulatory roles. Some examples include:

Line 250: "Overall, our findings highlight the complex role of PRKD3 as both a potential diagnostic and prognostic biomarker across diverse cancer types."

Line 353: "Collectively, these findings suggest that PRKD3 expression is intricately linked to multiple layers of immune regulation across cancers... potentially modulating tumor immunogenicity and influencing responsiveness to ICI therapy."

Line 379: "...these findings suggest that PRKD3 may influence tumor immune responsiveness through modulation of the tumor microenvironment and immune signaling pathways in syngeneic models."

Additionally, we reinforced the correlative nature of our findings rather than causal links: “Overall, our findings propose PRKD3 as a potential context-dependent biomarker to inform precision oncology; however, further validation studies are needed to establish mechanistic roles and clinical utility across diverse tumor types.”

2. AUC values from TCGA vs GTEx may be inflated due to uncorrected batch effects and lack of external validation datasets.

We thank the reviewer for highlighting these critical validation issues. To address potential batch effects inflating AUC values, ComBat correction was applied to TCGA-GTEx TOIL TPM data across cancer cohorts with sufficient normal samples, removing dataset-specific technical effects while preserving biological tumor/normal variation. In the revised Figure 2, we clearly distinguish batch-corrected from non-corrected ROC curves. Full ROC data is available in Supplementary Table S2. Additionally, we acknowledging limitations requiring external clinical validation in our discussion:

Line 417: "Given limited prior data on PRKD3 clinicopathological associations and expression discrepancies, further clinical studies across diverse patient cohorts and molecular subtypes required to validate the potential links reported in this study."

3. TISMO datasets are based on mouse syngeneic models, which differ from human tumors. Their relevance to clinical immunotherapy response should be more carefully qualified.

We thank the reviewer for emphasizing the translational limitations of mouse models. We have clarified throughout our manuscript that TISMO datasets derive from mouse syngeneic immunotherapy studies and qualified their potential role their clinical relevance in the Discussion. These revisions transparently frame the hypothesis-generating nature of these findings while calling for human validation studies

Line 484: "Although derived from mouse syngeneic models that differ from human tumors, these data suggest PRKD3 potentially modulates immune responsiveness through microenvironmental and signaling alterations, indicating cancer-specific predictive value as a potential immunotherapy biomarker that warrants further human clinical validation."

4. The concept of PRKD3 acting as a “paradox gene” is interesting, but conclusions require analyses that adjust for tumor purity, immune subtypes, or molecular subtypes.

We appreciate the reviewer’s insightful comments on the “paradox gene” concept. In the revised Discussion, we emphasized its speculative and hypothetic nature and explicitly highlights the need for a more in-depth molecular analysis.

Line 424: “To our knowle

---

## [Decision Letter · Decision Letter 1]

17 Mar 2026

Pan-Cancer Landscape of Protein Kinase D3: An Integrative TCGA Multi-Omics Analysis of Clinical, Molecular, and Immunological Roles

PONE-D-25-52894R1

Dear Dr. Leitges,

We’re pleased to inform you that your manuscript has been judged scientifically suitable for publication and will be formally accepted for publication once it meets all outstanding technical requirements.

Kind regards,

Amr Ahmed El-Arabey

Academic Editor

PLOS One

Additional Editor Comments (optional):

Reviewers' comments:

Reviewer's Responses to Questions

**Comments to the Author**

Reviewer #2: All comments have been addressed

2. Is the manuscript technically sound, and do the data support the conclusions?

Reviewer #2: Yes

3. Has the statistical analysis been performed appropriately and rigorously?

Reviewer #2: Yes

4. Have the authors made all data underlying the findings in their manuscript fully available?

Reviewer #2: Yes

5. Is the manuscript presented in an intelligible fashion and written in standard English?

Reviewer #2: Yes

Reviewer #2: The authors have satisfactorily addressed all concerns raised in the previous round of review. The revised manuscript shows clear improvements in methodological rigor, statistical transparency, and interpretative caution.

Key issues have been appropriately resolved, including batch-effect correction for TCGA–GTEx analyses, multivariate Cox regression adjusting for relevant clinical variables, and consistent application of false discovery rate correction. The consensus-based approach used to address discrepancies among immune deconvolution methods is appropriate for pan-cancer immune analyses. Causal language has been revised throughout, and interpretations related to immune regulation and immunotherapy response are now properly qualified.

The study employs suitable statistical methods and sufficiently large public datasets, and the conclusions are well aligned with the data presented without overstatement. No concerns regarding research ethics, data availability, or publication ethics were identified.

.

Reviewer #2: No

---

## [Editor Report · Acceptance letter]

PONE-D-25-52894R1

PLOS One

Dear Dr. Leitges,

I'm pleased to inform you that your manuscript has been deemed suitable for publication in PLOS One. Congratulations! Your manuscript is now being handed over to our production team.

Kind regards,

on behalf of

Dr. Amr Ahmed El-Arabey

Academic Editor

PLOS One